# 2′-Fucosyllactose Remits Colitis-Induced Liver Oxygen Stress through the Gut–Liver–Metabolites Axis

**DOI:** 10.3390/nu14194186

**Published:** 2022-10-08

**Authors:** Qianqian Yao, Yanan Gao, Linlin Fan, Jiaqi Wang, Nan Zheng

**Affiliations:** 1Key Laboratory of Quality & Safety Control for Milk and Dairy Products of Ministry of Agriculture and Rural Affairs, Institute of Animal Sciences, Chinese Academy of Agricultural Sciences, 2# Yuanmingyuan West Road, Haidian District, Beijing 100193, China; 2State Key Laboratory of Animal Nutrition, Institute of Animal Sciences, Chinese Academy of Agricultural Sciences, Beijing 100193, China; 3Gembloux Agro-Bio Tech, University of Liège, B-5030 Gembloux, Belgium

**Keywords:** 2′-fucosyllactose (2′-FL), liver oxygen stress, gut microbiota, glycometabolism, lipid metabolism, *Akkermansia muciniphila*, *Paraprevotella* spp.

## Abstract

Liver oxygen stress is one of the main extraintestinal manifestations of colitis and 5% of cases develop into a further liver injury and metabolic disease. 2′-fucosyllactose (2′-FL), a main member of human milk oligosaccharides (HMOs), has been found to exert efficient impacts on remitting colitis. However, whether 2′-FL exerts the function to alleviate colitis-induced liver injury and how 2′-FL influences the metabolism via regulating gut microbiota remain unknown. Herein, in our study, liver oxygen stress was measured by measuring liver weight and oxygen-stress-related indicators. Then, 16S full-length sequencing analysis and non-target metabolome in feces were performed to evaluate the overall responses of metabolites and intestinal bacteria after being treated with 2′-FL (400 mg/kg b.w.) in colitis mice. The results showed that, compared with the control group, the liver weight of colitis mice was significantly decreased by 18.30% (*p* < 0.05). After 2′-FL treatment, the liver weight was significantly increased by 12.65% compared with colitis mice (*p* < 0.05). Meanwhile, they exhibited higher levels of oxidation in liver tissue with decreasing total antioxidant capacity (T-AOC) (decreased by 17.15%) and glutathione (GSH) levels (dropped by 22.68%) and an increasing malondialdehyde (MDA) level (increased by 36.24%), and 2′-FL treatment could reverse those tendencies. Full-length 16S rRNA sequencing revealed that there were 39 species/genera differentially enriched in the control, dextran sulphate sodium (DSS), and DSS + 2′-FL groups. After treatment with 2′-FL, the intestinal metabolic patterns, especially glycometabolism and the lipid-metabolism-related process, in DSS mice were strikingly altered with 33 metabolites significantly down-regulated and 26 metabolites up-regulated. Further analysis found DSS induced a 40.01%, 41.12%, 43.81%, and 39.86% decline in acetic acid, propionic acid, butyric acid, and total short chain fatty acids (SCFAs) in colitis mice (all *p* < 0.05), respectively, while these were up-regulated to different degrees in the DSS + 2′-FL group. By co-analyzing the data of gut microbiota and metabolites, glycometabolism and lipid-metabolism-associated metabolites exhibited strong positive/negative relationships with *Akkermansia_muciniphila* (all *p* < 0.01) and *Paraprevotella* spp. (all *p* < 0.01), suggesting that the two species might play crucial roles in the process of 2′-FL alleviating colitis-induced liver oxygen stress. In conclusion, in the gut–liver–microbiotas axis, 2′-FL mediated in glucose and lipid-related metabolism and alleviated liver oxygen stress via regulating gut microbiota in the DSS-induced colitis model. The above results provide a new perspective to understand the probiotic function of 2′-FL.

## 1. Introduction

Inflammatory bowel disease (IBD) is a chronic, recurring inflammatory response involving gut microbiota dysbiosis and metabolic syndrome [1]. Several studies have shown that liver injury is a feature of colitis patients, with 5% of them developing a further liver injury and metabolic disease [2,3]. Rojas-Feria et al. (2013) found that 30% of IBD patients exhibited an abnormal liver biochemistry [4]. Studies showed that the shared pathogenic components that contribute to the consistency of liver damage and IBD may be oxidative stress, dysbiosis of gut microbiota, and inflammation [5,6]. In view of the high exposure of bacterial toxins for the liver, gut microbiota dysbiosis in colitis may be involved in liver injury via the gut–liver axis. When the gut barrier is damaged, bacteria-derived metabolites and toxins are transferred to the liver, which could then induce various liver diseases [7]. Oxidative stress is another vital factor for liver injury, which may induce the disorder of various metabolisms, including lipid, carbohydrate, and trace elements, and consequently influencing the health of the host. Compared with signal microbial compositions or metabolomics analysis, their co-analysis can reveal the key metabolites of intestinal microbiome in response to host pathophysiology and provide clues for the study of the mechanism of the microbiome–host interaction. Franzosa et al. (2019) co-analyzed microbial sequencing and metabolomics data to explore the association between metabolites and bacterial flora, and found that the eicosatrienoic acid (ETA ) enriched in IBD was negatively correlated with the health-related species *Eubacterium ventriosum* [8]. These findings offered a theoretical framework for comprehending the root causes of IBD and locating prospective treatment targets.

As the standard nutrition source, human milk is of significance to infants in their early life. Notably, breastfeeding is related to overt benefits to the infants, including decreasing risks of necrotizing enterocolitis (NEC), leukemia, obesity, and type 1 and 2 diabetes, among others [9,10]. Oligosaccharides is a unique component in mammal milk, not found in infant formula, and that in human milk is 100- or 1000-fold more than that in bovine milk [11,12]. After lactose and milk fat, oligosaccharides make up the third-largest component of human milk, with quantities ranging from 5–12 g/L in mature human milk to >20 g/L in colostrum [11,13]. Ongoing research discovers that human milk oligosaccharides (HMOs) play important roles in improving the development of infants through enhancing the immune system, binding pathogens, and interacting with multiple epithelial cells to improve intestinal barrier function [14,15].

The most prevalent HMO is 2′-fucosyllactose (2′-FL), with concentrations ranging from 0.026 to 4.5 g/L [16,17]. It can be fermented by *Lactobacillus*, *Bifidobacterium*, and *Bacteroides* in the colon and produce lactic acid, acetic acid, and 1, 2-propylene glycol, which then participate in various metabolic pathways [18,19]. It confers short- and long-term benefits as a prebiotic by degrading by gut bacteria to promote probiotics’ growth [20] and preventing the colonization of pathogenic bacteria [21]. More importantly, it can efficiently suppress inflammation. Holscher et al. (2014) assessed the influence of 2′-FL on intestinal cells and found that it could suppress the proliferation of intestinal cells and promote the differentiation of Caco-2Bbe and HT-29 cell lines *in vitro* [22]. He et al. (2016) observed that 2′-FL reduced the expression of the lipopolysaccharide (LPS) receptor (CD14) on intestinal epithelial cells and inhibited the release of pro-inflammatory signaling molecules, reversing the inflammatory response induced by *Escherichia coli* infection of intestinal epithelial cells [23]. In several in vivo experiments, 2′-FL showed the ability to alleviate the inflammation in the colitis model by regulating gut microbiota composition and improving mucin secretion [24,25].

In our previous study, we found that 2′-FL could ameliorate colitis via regulating gut microbiota and stimulating mucin secretion; however, whether 2′-FL exerts the function to alleviate colitis-induced liver injury and how 2′-FL influences the metabolism via regulating gut microbiota remain unknown. In this study, we attempt to untangle some aspects of this tripartite microbiota–liver-metabolites crosstalk and, consequently, attempt to assemble meaningful and applicable conclusions, which will provide a new perspective to understand the probiotic function of 2′-FL.

## 2. Materials and Method

### 2.1. Chemicals

2′-FL (purity ≥ 98%, cat# GY1141) was purchased from HuicH Technology Co., Ltd. (Shanghai, Beijing, China). Glutathione (GSH, cat: BC1170), malondialdehyde (MDA, cat: BC0020), total antioxidant capacity (T-AOC, cat: BC1310), and superoxide dismutase (SOD, cat: BC0170) were bought from Solarbio Life Science (Beijing, China).

### 2.2. Animals Experiment

Eighteen C57BL/6J male mice (18–22 g) were divided into three groups: the control group, the dextran sulphate sodium (DSS) group, and the DSS + 2′-FL group, as described in a prior work [25]. Mice in the DSS + 2′-FL groups received 0.3 mL of 400 mg/kg b.w. of 2′-FL orally once a day from 0 to 21 days. Equal amounts of phosphate buffer solution (PBS) were given to the mice in the control group and DSS group at the same time. To create a colitis model, mice in the DSS groups were given 5% (*w*/*v*) DSS in their water ad libitum for 7 consecutive days between 14 and 21 days. The Chinese Academy of Agriculture Sciences’ Committee on the Ethics of Animal Experiments accepted the study’s procedure (Beijing, China; permission number: IAS-2021-03).

### 2.3. Oxidation Index Detection

Liver tissue (0.1 g) was accurately weighed and then 1 mL extracting solution was added. The tissue was ground thoroughly using an electric pestle and centrifuged at 8000× *g* for 10 min. The supernatant was taken to be measured. GSH, MDA, T-AOC, and SOD were detected according to manufacturer’s operation manual. The serum was also to be collected to detect all of the above indexes.

### 2.4. DNA Extraction and Full-Length 16S rRNA Sequencing

The full-length 16S rRNA gene was amplified using the fecal DNA (*n* = 6 for each group) and the primers 27F: AGRGTTTGATYNTGGCTCAG and 1492R: TASGGHTACCTTGTTASGACTT. Lima (v 1.7.0) and the UCHIME algorithm were used to examine the qualified circular consensus sequencing (CCS) reads after the raw data had been filtered (v 8.1). Substandard segments (those without primers and longer than 1200–1650 bp) and chimeric sequences were eliminated to produce clean readings. Using USEARCH (Edgar, 2013) (v 10.0), sequences with similarity 97% were grouped into the same operational taxonomic unit (OTU). The BMK Cloud platform (https://international.biocloud.net/zh/dashboard, accessed on 23 March 2021) served as the platform for the bioinformatics analysis of this study.

### 2.5. Non-Target Metabolome in Feces

The supernatant fecal samples for LC-MS/MS were prepared as previous described [26]. Here, 50 mg feces was homogenated with 500 μL of ice-cold methanol/water (70%, *v*/*v*), then it was centrifuged at 12,000× *g* rpm at 4 °C for 10 min and the supernatant was taken to concentrate. Then, 100 μL of 70% methanol water was added into the dried product and ultrasonic treatment was performed for 3 min, followed by centrifuging at 12,000× *g* rpm at 4 °C for 3 min. Then, 60 μL of supernatant was sucked for LC-MS/MS analysis, with six replicates for each group.

The analytical conditions were as follows: UPLC: column, Waters ACQUITY UPLC HSS T3 C18 (1.8 µm, 2.1 mm × 100 mm); column temperature, 35 °C; flow rate, 0.3 mL/min; injection volume, 1 μL; solvent system, water (0.01% methanolic acid): acetonitrile; gradient program of positive ion, 95:5 *v*/*v* at 0 min, 79:21 *v*/*v* at 3.0 min, 50:50 *v*/*v* at 5.0 min, 30:70 *v*/*v* at 9.0 min, 5:95 *v*/*v* at 10.0 min, and 95:5 *v*/*v* at 14.0 min; gradient program of negative ion, 95:5 *v*/*v* at 0 min, 79:21 *v*/*v* at 3.0 min, 50:50 *v*/*v* at 5.0 min, 30:70 *v*/*v* at 9.0 min, 5:95 *v*/*v* at 10.0 min, and 95:5 *v*/*v* at 14.0 min. After data quality control, metabolic identification information was obtained by searching the laboratory’s self-built database and integrating the public database and metDNA. Finally, statistical analysis was carried out by R program. Statistical analysis includes univariate analysis and multivariate analysis.

### 2.6. Short-Chain Fatty Acids (SCFAs) Content

Here, 0.2 g colon contents were homogenized with 1 mL ultra-pure water, and then 250 uL of 25% metaphosphoric acid solution was added. The solution was kept in an ice bath for 30 min and centrifuged at 5000× *g* rpm for 10 min (4 °C), then the pH was adjusted to 2–3. After standing for 5 min, the supernatant was filtered by a 0.22 μm filter and tested on gas chromatography equipped with a hydrogen flame ionization detector (7890 A, Agilent Technologies, Santa Clara, CA, USA). SCFAs standards were purchased from Alta Scientific. Ltd. (Tianjin, China).

### 2.7. Statistical Analysis

The data were presented as mean and standard error of the mean (SEM). The differences between groups were compared using Dunnett’s *t*-test multiple comparisons in SPSS 19.0. Statistical significance was defined as *p* < 0.05. The aforementioned data were used to create charts using Graphpad Prism 9.0.

## 3. Results

### 3.1. 2′-FL Alleviated Colitis-Induced Oxidative Stress in the Liver

Liver injury has been recognized to be one of the main extraintestinal manifestations of colitis [27]. In our study, we found that the weight of the liver in the DSS group was significantly lower compared with the control group (Figure 1A,B, *p* < 0.05), and this decline was reversed by 2′-FL. The ratio of liver weight to body weight had no significant difference among the three groups (Figure 1C, *p* > 0.05). The levels of T-AOC and GSH in liver were remarkably increased, and MDA in the liver was significantly decreased after being treated with 2′-FL when compared with colitis mice (Figure 1D, *p* < 0.05). Correspondingly, compared with the DSS group, the concentration of SOD in serum was increased distinctly and MDA in serum was significantly decreased after 2′-FL treatment (Figure 1E, *p* < 0.05), suggesting systemic oxidative stress. All of the above data revealed that colitis induced the oxidative stress in the liver, and 2′-FL could effectively alleviate it.

### 3.2. 2′-FL Altered the Profile of Gut Microbiota in Colitis Mice

The full-length 16S rRNA sequencing of the fecal microbiota was then carried out to determine the unique alterations in the makeup of the gut microbiota following the addition of 2′-FL under conditions of liver injury. To identify the species that 2′-FL artificially altered, the relative abundance of microbes was compared across groups, and the *p*-value was determined using ANOVA. The results showed that, viewed collectively, the picture in the DSS + 2′-FL group was strikingly different from that in the DSS group, as well as that in the control group (Figure 2A), with 39 species/genera differentially enriched in the three groups (*p* < 0.05). Specifically, nine of them are further shown in Figure 2B. *Lachnospiraceae_NK4A136*, *Iiebacterium_valenns*, *Lactobacillus_reuteri*, and *Paraprevotella* spp. were relatively over-represented in individuals with cotilis (*p* < 0.05), while *Bifidobacterium_animalis*, *Akkermansia_muciniphila*, and *Bacteroides_caecimuris* were relatively enriched in the control and DSS + 2′-FL groups (*p* < 0.05).

### 3.3. 2′-FL-Associated Metabolic Patterns

Gut microbiota can affect the metabolism of nutrients, such as carbohydrates, lipids, and amino acids, so alteration in the composition of gut microbiota may induce changes in metabolic patterns. To further figure out the changes in metabolic patterns’ response to 2′-FL, non-target metabolomics in fecal was performed by LC-MS/MS. An apparent cluster separation was shown among each of the two groups in the orthogonal projections to latent structures-discriminant analysis (OPLS-DA) analysis (Figure 3A), indicating that the metabolic patterns in those groups were strikingly altered. Regarding the control and DSS groups, there are 89 significantly down-regulated metabolites and 100 up-regulated ones (Figure 3B). Those significantly changed metabolites were enriched in tryptophan metabolism, valine, leucine, and isoleucine biosynthesis, as well as inflammatory mediator regulation of TRP channels. As for the DSS and DSS + 2′-FL groups, 33 metabolites were significantly down-regulated and 26 ones were up-regulated (Figure 3B). In detail, those significantly changed metabolites were enriched in the degradation of aromatic compounds, valine, leucine, and isoleucine biosynthesis, as well as biosynthesis of amino acids. It is noteworthy that many metabolites were found to group to glycometabolism-related pathways, such as glycolysis/gluconeogenesis, fructose and mannose metabolism, and mannose type O-glycan biosynthesis.

### 3.4. 2′-FL Altered the Glycometabolism in Colitis Mice

2′-FL, a main type of HMO, cannot be digested in the upper digestive tract because of the lack of glycosidehydrolase. When arriving at the colon, it will be fermented by gut microbiota, and may subsequently induce a series of changes in glycometabolism. Therefore, we further analyzed the metabolites associated with carbohydrate metabolism and 15 remarkably changed ones were screened out. As shown in Figure 4, among them, nine metabolites in the DSS group (fructose-lysine, D-galactosamine, β-D-lactose, D-maltose, galactosylsphingosine, N-acetyputrescine, ergosteryl 3-β-D-glucoside, 2-aminogalactopyranose, and fructoselysine-6-phosphate) declined significantly when compared with the control group (*p* < 0.05) and were slightly up-regulated after 2′-FL treatment. The remaining six metabolites were increased in the DSS group, i.e., maltopentaose (*p* < 0.05), D-fructuronate (*p* < 0.05), N-Succinyl-2-amino-6-ketopimelate (*p* > 0.05), octyl alpha-D-glucopyranoside (*p* < 0.05), UDP-3-O-(3-hydroxymyristoyl)-N-acetylglucosamine (*p* > 0.05), and fructose 1,6-bisphosphate (*p* > 0.05). 2′-FL treatment significantly enhanced the N-Succinyl-2-amino-6-ketopimelate (*p* < 0.05), D-fructuronate (*p* = 0.0659), and 2-aminogalactopyranose (*p* < 0.05) abundance.

### 3.5. 2′-FL Influenced the Lipid Metabolism in Colitis Mice

Gut microbiota can regulate the absorption, storage, and energy acquisition of the host, thus participating in lipid metabolism. Thus, we analyzed the lipid metabolism and found 12 significantly altered metabolites. Among them, the levels of phosphatidyl ethanolamine (PE, 18:0), PE (16:0/14:0), PE (14:0/16:0), lysophosphatidyl choline (LysoPC, 18:2), phosphatidyl choline (PC, 18:1), and phosphatidyl inositol (PI, 18:0/18:1) in the DSS group were significantly increased compared with the control (Figure 5, *p* < 0.05). After being treated with 2′-FL, PE (18:0) and PE (16:1) were remarkably declined, while the levels of PG (16:1/18:1), PI (16:1), LysoPE (20:3), and PE (18:1) were significantly up-regulated (Figure 5, *p* < 0.05). We also detected changes in SCFAs in feces using GC-MS. The results showed that DSS induced a sharp decline in acetic acid, propionic acid, and butyric acid, as well as in total SCFAs in colitis mice (*p* < 0.05), while they were up-regulated to different degrees in the DSS + 2′-FL group. There was no significant difference in the concentrations of isobutyric acid, valeric acid, and isovaleric acid among the three groups (*p* > 0.05).

### 3.6. The Correlation Analysis between Specific Microbes and Metabolites

To further understand the relationship between metabolism and the microenvironment, the correlation heatmap of microbes and metabolites was created. As Figure 6A shows, several strong positive/negative relationships were found among microbes and specific metabolites. Among them, *Akkermansia_muciniphila* and *Paraprevotella* spp. were the closest relevant to those glycometabolism and lipid-related mebabolites. In detail (Figure 6B), *Akkermansia_muciniphila*, a significantly down-regulated species in the DSS group, were positively correlated with propionic acid (R^2^ = 0.4186, *p* = 0.0037), fructoselysine (R^2^ = 0.7324, *p* < 0.0001), 7,8-Dihydroionone (R^2^ = 0.63, *p* < 0.0001), PE (16:0) (R^2^ = 0.4910, *p* = 0.0012), and D-maltose (R^2^ = 0.6291, *p* < 0.0001), and negatively correlated with D-2-aminopentanoic acid (R^2^ = 0.5269, *p* = 0.0006), LysoPC (18:2(9Z,12Z)) (R^2^ = 0.3762, *p* = 0.0068) and PE (16:0/14:0) (R^2^ = 0.3568, *p* = 0.0088), PC (18:1(9Z)) (R^2^ = 0.3762, *p* = 0.0068), and 20-hydroxy-PGF2a (R^2^ = 0.4168, *p* = 0.0037). However, the relationship between metabolites and *Paraprevotella* spp. was opposite to that with *Akkermansia_muciniphila*. Specifically, *Paraprevotella* spp. was negatively correlated with propionic acid (R2 = 0.2567, *p* = 0.0319), fructoselysine (R^2^ = 0.5558, *p* = 0.0004), 7,8-Dihydroionone (R^2^ = 0.4935, *p* = 0.0012), PE (16:0/0:0) (R^2^ = 0.5624, *p* = 0.0003), and D-maltose (R^2^ = 0.6291, *p* < 0.0001), and positively correlated with D-2-aminopentanoic acid (R^2^ = 0.6512, *p* < 0.0001), LysoPC (18:2(9Z,12Z)) (R^2^ = 0.4748, *p* = 0.0016) and PE (16:0/14:0) (R^2^ = 0.4748, *p* = 0.0016), PC (18:1(9Z))(R^2^ = 0.5322, *p* = 0.0006), and 20-hydroxy-PGF2a (R^2^ = 0.6365, *p* < 0.0001). These results suggested that the two species might play crucial roles in glycometabolism and lipid-related metabolism, and thus are involved in the process of 2′-FL alleviating colitis-induced liver oxygen stress.

## 4. Discussion

To untangle the crosstalk of the gut–liver axis in the process of 2′-FL ameliorating colitis-induced liver injury, we firstly compared the physical condition and oxygen stress of liver in colitis mice with or without 2′-FL treatment. Then, we examined the changes in gut microbiota and metabolites, respectively, and then the glucose and lipid-related metabolites were analyzed to locate the specific bacteria and metabolites’ response to 2′-FL in the condition of liver oxygen stress. We further investigated the correlation relationship between the differential microbes and metabolites to uncover the links between 2′-FL, bacterial, and metabolites in colitis-induced liver damage mice. Moreover, we found that, in the gut–liver–microbiotas axis, 2′-FL mediated glucose and lipid-related metabolism and alleviated liver oxygen stress via regulating gut microbiota in the DSS-induced colitis model.

As an important metabolic organ, the liver is responsible for the metabolism, synthesis, detoxification, and excretion of compounds.

Several studies have found that liver injury is one of the main extraintestinal manifestations of colitis [2,27]. The excess of reactive oxygen species (ROS) metabolites in the intestinal mucosa of colitis patients will induce oxygen stress and then cause liver injury [28,29]. In the present study, we examined oxidative stress in the liver. As expected, a significant liver weight loss and an obvious increase in the oxygen stress index were observed in the DSS group. 2′-FL was found to exert antioxidant functions through protecting liver physiological function, i.e., increasing T-AOC and GSH in the liver and decreasing MDA in serum.

HMOs are largely digested by intestinal enzymes and gut microbiota [30]. Low intact HMOs are found in feces, urine, and plasma, which have passed through the intestine and been absorbed into the blood system [31,32]. Goehring et al. (2014) reported the presence of HMO, including 2′-FL, in the urine and plasma of breastfed, but not in formula-fed, infants, and 2′-FL was not present in the blood system in infants fed with non-2′-FL breast milk [33]. Most of 2′-FL is selectively fermented and utilized by gut microbiota to further exert beneficial functions. In vitro fermentation experiments showed that *Bifidobacterium* and *Bacteroides* could degrade more than 40% of 2′-FL, while the consumption rate of Clostridium, *Staphylococcus*, and *Escherichia coli* was less than 10% [20,34]. Most of the *Bifidobacteria* in the intestinal tract is believed to specifically express fucosidases to cleave Fuc and degrade HMOs. *Bifidobacterium Longum* subsp. *Infantis* and *Bifidobacterium infantis* showed good growth on the medium with 2′-FL as the only carbon source with ABC sugar transporter-related gene clusters up-regulated during *Bifidobacterium infantis* metabolism of 2′-FL [34,35]. The metabolites produced by 2′-FL in the gut include lactate, formate, acetate, fucose, and 1, 2-propanediol [18,19]. In our present study, we found that the composition of gut microbiota was strikingly altered by 2′-FL, with 39 species/genera differentially enriched among the three groups. Specifically, *Lachnospiraceae_NK4A136*, *Rumincoccaceae_UCG-014*, and Paraprevotella.spp relatively were relatively over-represented in individuals with cotilis, while *Bifidobacterium_animalis*, *Akkermansia_muciniphila*, and *Bacteroides_caecimuris* were relatively enriched in the control and DSS + 2′-FL groups. Besides that, those supplemented with 2′-FL also increased the abundance of *Prevotellaceae* (4.3% in the DSS group vs. 5.5% in the DSS + 2FL group), which has been found to be associated with complex carbohydrate intake [36]. All of those results suggested that the intestinal flora adapted to a diet rich in 2′-FL enables the host to efficiently extract energy, thus preventing inflammation.

As for metabolism, the metabolism pattern in colitis mice was significantly altered by DSS compared with the control group, with 89 significantly down-regulated metabolites and 100 up-regulated ones. It is worth noting that the differential metabolites in the 2′-FL-supplemented group clustered to glycometabolism-related pathways, such as glycolysis/gluconeogenesis, fructose and mannose metabolism, and mannose type O-glycan biosynthesis, indicating that the glycometabolism in colitis mice might be influenced by 2′-FL. Glucose metabolic disorders have been shown to be involved in inflammation [37,38]. To elucidate this, we further analyzed the metabolites associated with carbohydrate metabolism, and 15 remarkably changed ones were screened out. Among them, D-fructuronate is an isomer of D-glucuronate catalyzed by uronate isomerase from gut bacteria (*Escherichia coli*, *Bacillus amyloliquefaciens* group) in the first step of the utilization of d-glucuronic acid, and thus is involved in the sugar’s hydrolysis reaction [39,40]. Fucose is a necessary glycan residue that is present on the surface of cells and is needed for a number of biological functions, including inflammation, self-recognition, and cell-to-cell communication [41]. The two primary mechanisms for controlling glucose metabolism are gluconeogenesis and glycogenolysis. By converting lactate, glycerol, and glycogenic amino acids into glucose or by converting glycogen into glucose, fructose-1,6-bisphosphatase and glucose 6-phosphatase participate in the gluconeogenesis pathway. Fructose-1,6-bisphosphatase inhibitors and decreased glucose 6-phosphatase gene expression were both shown to lower blood glucose levels in type 2 diabetes in earlier research [42,43].

Chronic inflammation can also cause the disorder of lipid metabolism [44]. Lipid metabolism plays an important role in maintaining multiple biochemical and immune functions. Phosphatidyl choline (PC) and phosphatidyl ethanolamine (PE) have been implicated in steatosis and phosphatidyl liposis [45]. LysoPCs, negatively correlated with PCs, can suppress the PCs’ formation [46]. LysoPE, a neurotrophic activator, can increase intracellular calcium levels and stimulate chemotactic migration and cell invasion in cancer [47]. In our study, colitis induced an increase in PE (18:0), PE (16:0/14:0), PE (14:0/16:1), LysoPC (18:2), PC (18:1), and phosphatidyl inositol (PI, 18:0/18:1). However, 2′-FL decreased the levels of PE (18:0) and PE (16:1), and increased phosphatidyl glycerol (PG 16:1/18:1), PI (16:1), LysoPE (20:3), and PE (18:1) levels. We further investigated the correlation relationship between the differential microbes and metabolites, and found that *Akkermansia_muciniphila* and *Paraprevotella*.spp exhibited the closest relevance to those glycometabolism mebabolites. In detail, *Akkermansia_muciniphila* was positively correlated with Beta-D-lactose, D-maltose, fructoselysine, and D-galactosamine, and negatively correlated with D-2-aminopentanoic acid and D-fructuronate. However, the relationship between metabolites and *Paraprevotella.spp* was opposite to that with *Akkermansia_muciniphila*. These results suggested that the two species might play crucial roles in glycometabolism, and are thus involved in the process of 2′-FL alleviating colitis-induced damage. However, 16S RNA sequencing and metabolomics are a relatively quantitative approach, and additional PCR and other analytical techniques are required to precisely quantify the changes in bacterial flora and metabolites. Additionally, our future work will also concentrate on employing in vivo and in vitro models to better understand the association between the crucial metabolites screened by our current study and colitis-induced liver injury.

## 5. Conclusions

In the mice model of colitis-induced liver injury, 2′-FL therapy increased GSH and T-AOC levels by 20.02% and 13.31%, respectively, and increased liver weight by 12.64% in comparison with the DSS group. Furthermore, 2′-FL mediated glucose and lipid-related metabolism and alleviated liver oxygen stress through the gut–liver–microbiotas axis. The above results provide a new perspective to understand the probiotic function of 2′-FL.

## Figures and Tables

**Figure 1 nutrients-14-04186-f001:**
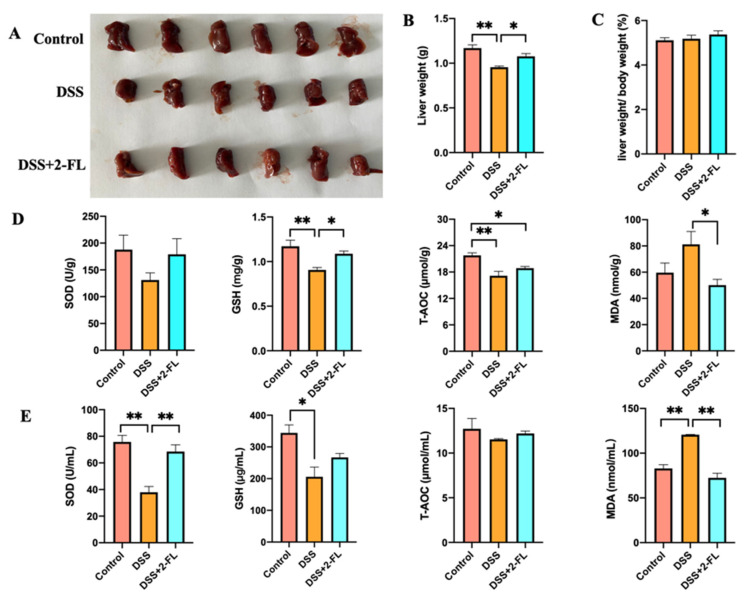
2′-FL alleviated colitis-induced oxidative damages in the liver. (**A**) The liver of individual mice in three groups. (**B**) Liver weight. (**C**) The ratio of liver weight to body weight. (**D**) The levels of Glutathione (GSH), malondialdehyde (MDA), total antioxidant capacity (T-AOC) and superoxide dismutase (SOD) in the liver (*n* = 6). (**E**) The levels of T-AOC, GSH, MDA, and SOD in serum (*n* = 4). Significance determined using one-way ANOVA and expressed as mean ± SEM. *, *p* < 0.05. **, *p* < 0.01.

**Figure 2 nutrients-14-04186-f002:**
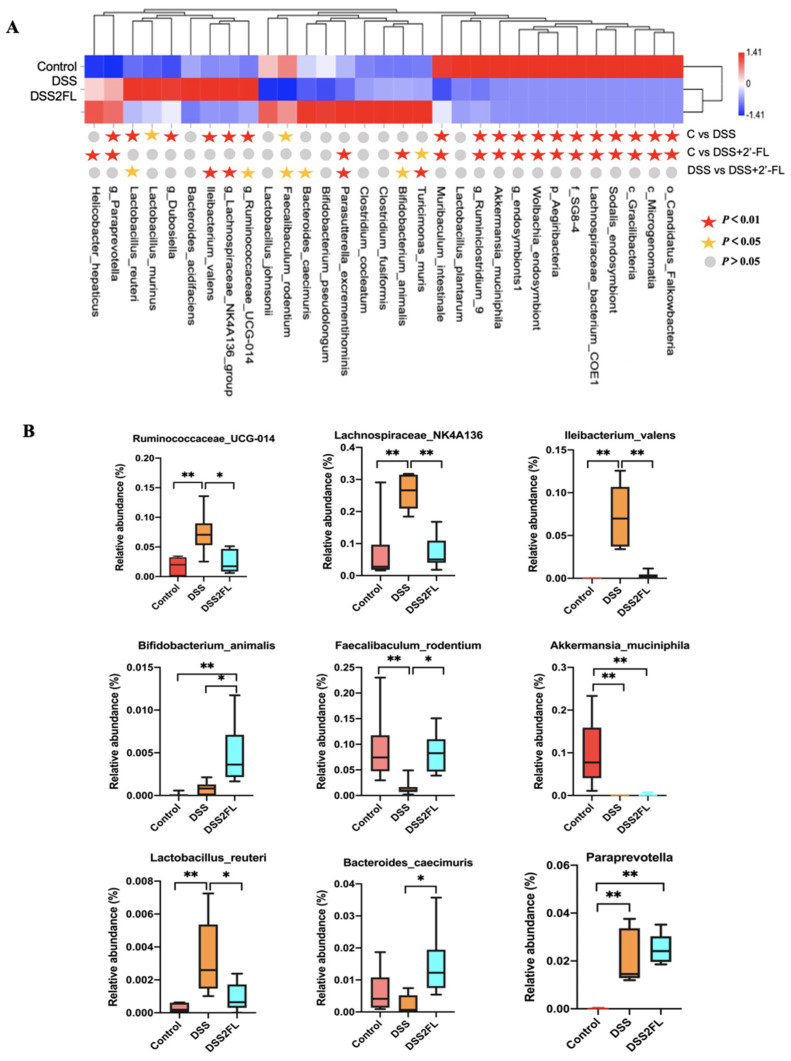
Strikingly differential species among the three groups. (**A**) The top 39 species across groups at the criteria of *p*-value < 0.05 by ANOVA rank sum test. The abundance profiles are transformed into Z scores by subtracting the average abundance and dividing the standard deviation of all samples. The Z score is negative (shown in blue) when the row abundance is lower than the mean. (**B**) The representative figures of significantly changed species across groups analyzed by one-way ANOVA test (*n* = 6 for each group). *, *p* < 0.05. **, *p* < 0.01.

**Figure 3 nutrients-14-04186-f003:**
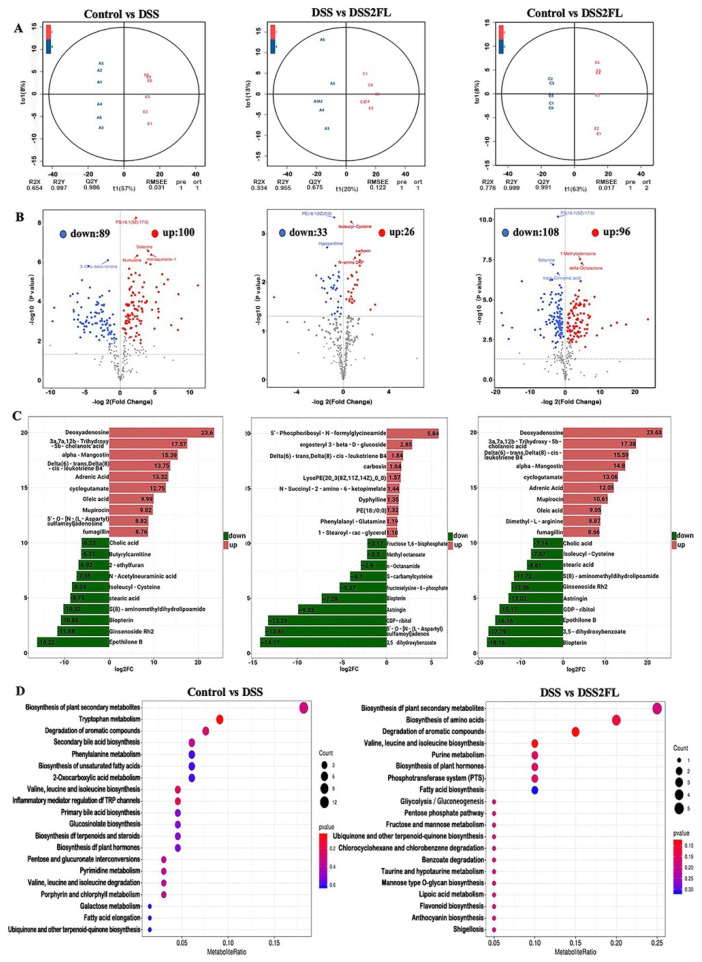
2′-FL-associated metabolic patterns in C57BL/6J mice. (**A**) The orthogonal projections to latent structures-discriminant analysis (OPLS-DA) among every two groups. (**B**) Volcano plot among every two groups with the criteria of fold-change (FC) threshold = 2 and *p* < 0.05. (**C**) The histogram of the enriched pathway of differential metabolites in B. (**D**) KEGG-enriched map of differential metabolites. In each panel, A, DSS group; C, DSS2FL group; and E, control group; *n* = 6 for each group.

**Figure 4 nutrients-14-04186-f004:**
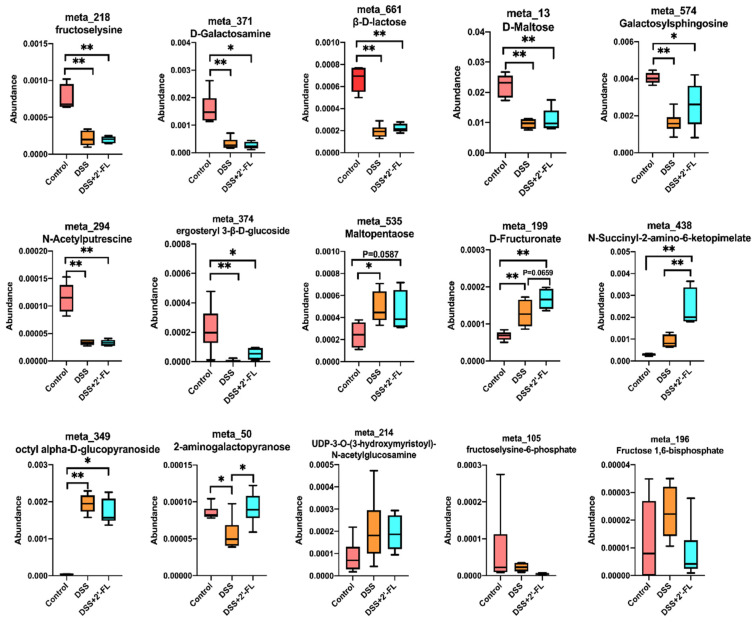
The profile of glycometabolism was significantly altered among the three groups. Significance determined using one-way ANOVA (*n* = 6 for each group). *, *p* < 0.05. **, *p* < 0.01.

**Figure 5 nutrients-14-04186-f005:**
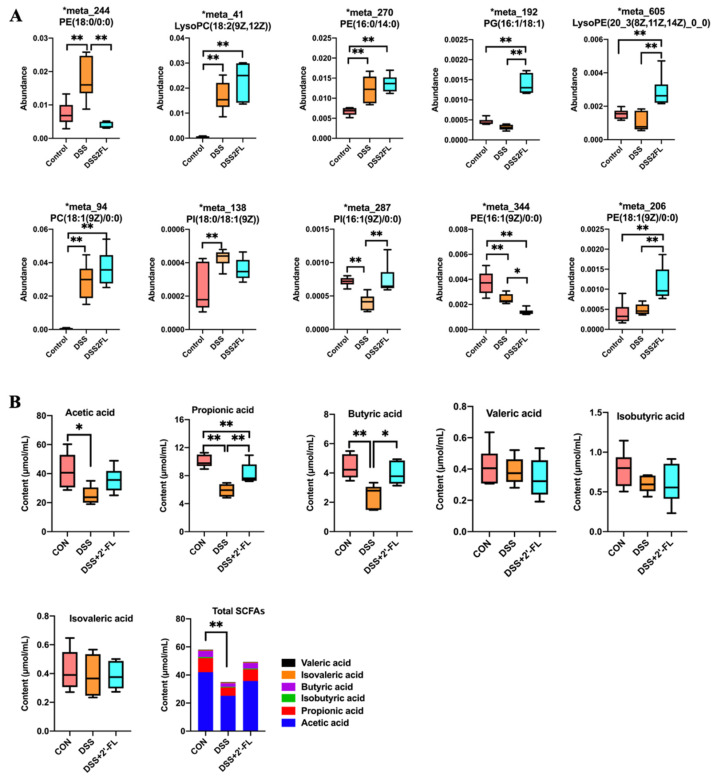
The profile of lipid metabolism and SCFAs was significantly influenced among the three groups. (**A**) The levels of metabolites related to lipid metabolism; (**B**) The contents of SCFAs. Significance was determined using one-way ANOVA (*n* = 6 for each group). *, *p* < 0.05. **, *p* < 0.01. PE, phosphatidyl ethanolamine; PI, phosphatidyl inositol; PC, phosphatidyl choline; PG, phosphatidyl glycerol; LysoPC, lysophosphatidyl choline; LysoPE, lysophosphatidyl ethanolamine.

**Figure 6 nutrients-14-04186-f006:**
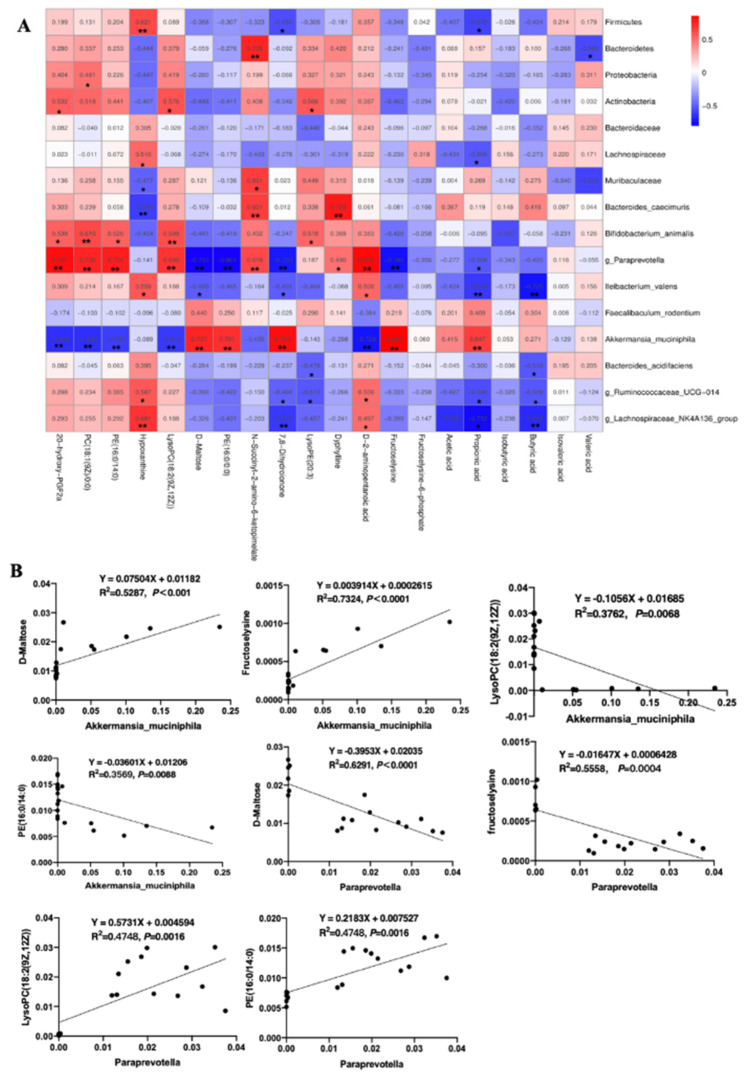
The correlation analysis between specific microbes and metabolites. (**A**) Correlation heatmap of microbes and metabolites in the colon (top 20). **, *p* < 0.01, *, *p* < 0.05. (**B**) The simple linear regression of representative microbes, mainly *Akkermansia_muciniphila* and *Paraprevotella* spp., and metabolites analyzed by Prism 9.0.

## Data Availability

16S amplicon data are deposited in the NCBI repository, accession number: BioProject PRJNA794307.

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
