# Peer review of "2′-Fucosyllactose Remits Colitis-Induced Liver Oxygen Stress through the Gut–Liver–Metabolites Axis"

_nutrients, 2022, doi:10.3390/nu14194186_

Round 1
Reviewer 1 Report
The paper by Yao Q., and Gao Y et al. aimed to describe the effect of 2'-fucosyllactose (2'-FL) on gut microbiota and metabolites and in consequence on liver oxygen stress in DSS induced colitis mice. In my opinion, the topic of the study undertaken by the authors is interesting and innovative, and the obtained results constitute the basis for further clinical trials. The manuscript is properly organized, and the datas are carefully described. However, I have few suggestions - included in the points under the appropriate section.
Introduction:
- The introduction is overall well written. However, the information about influence of IBD on gut microbiota and gut-derived metabolites and in consequence on liver injury development (not only by oxidative stress) are not enough. Please provide some more information about gut-liver microbiotas axis – in Introduction or in Discusion (page 12, after 300-304 lines)
The Material and Methods are well described.
Results:
2. The result of the review is sufficiently described, but the Figure 3 is not readable /very small font/
3. Please provide abbreviations in Table 5 footer (PE, PI, PC).
Discussion and conclusion
4. The discussion is based on the results, but how about the study limitations and future perspectives? Authors should more detailed explain the relationship between obtained results and liver injury to highlight the importance of the presented study.
Additional suggestion:
- Page 2. Line 54 – replace “sugar” on “carbohydrate”
- Page 2, line 61 – maybe “component” instead of “portion”
- Page 2, line 77 - the abbreviation” LPS” has not been explained before.
- Page 2, line 86 to 96 -> I suggest to move this fragment to the first paragraph that the authors describe that IBD influencing gut microbiota
- Page 3, line 111 - the abbreviation” PBS” has not been explained before.
- Page 4, line 172 – missing space
- The same situation for example page 9 line 248 (PE, PI, PC) – Authors explained these acronyms later in Discusion. Please check and correct the abbreviations use consistently throughout the manuscript.
Author Response
Thank you very much for the admiration and your comments on the work. We have revised our paper point by point with the comments of the reviewer which we hope to meet with approval. Further, the necessary corrections have been included for the improvement of the revised manuscript. The main corrections in the paper and the responds to the reviewer’s comments are attached below. Please see the attachment.

Reviewer 2 Report
The manuscript is very well written, is clear and has novelty.
There are some comments:
-The authors should mention the percentage. (line 24);
-the name of the microorganisms should be italicized (line 34, 69)
- should be added spp. after Paraprevotella (line 35)
- the abstract should contain more values of the results... increase/decrease how many times compared to the control
-the conclusion section should contain some data (values) that were obtained.
Author Response
Response to Reviewer 2 Comments
The manuscript is very well written, is clear and has novelty.
AU: Thanks to the reviewer for the admiration and your comments on the work. We have revised our paper point by point with the comments of the reviewer which we hope to meet with approval. Further, the necessary corrections have been included for the improvement of the revised manuscript. The main corrections in the paper and the responds to the reviewer’s comments are as following:
There are some comments:
- The authors should mention the percentage. (line 24);
AU: Thanks very much for your suggestion, and we have added the data in line 24, and that is showed as:
“The results showed that the compared with control group, liver weight of colitis mice was significantly decreased by 18.30% (P<0.05). After 2'-FL treatment, liver weight was significantly increased by 12.65% compared with colitis mice (P<0.05). Meanwhile, they exhibited higher levels of oxidation in liver tissue with decreasing T-AOC (decreased by 17.15%) and GSH levels (dropped by 22.68%) and increasing MDA level (increased by 36.24%) and 2'-FL treatment could reverse those tendency.”
“Further analysis found DSS induced a 40.01%, 41.12% 43.81% and 39.86% decline of acetic acid, propionic acid, butyric acid and total SCFAs in colitis mice (all P<0.05), while those were up-regulated in different degree in DSS+2'-FL group.”
Please see in lines 23-28, 32-35.
- the name of the microorganisms should be italicized (line 34, 69)
AU: Thank you for careful review. We carefully check through the manuscript and make all the name of the microorganisms italicized.
- should be added spp. after Paraprevotella (line 35)
AU: Thank you for careful review. We carefully check through the manuscript and replace “Paraprevotella” on “Paraprevotella. spp”.
- the abstract should contain more values of the results... increase/decrease how many times compared to the control
AU: Thanks very much for your suggestion, and we have added the relevant data in the abstract.
- the conclusion section should contain some data (values) that were obtained
AU: Thanks very much for your suggestion, and we have added the relevant data in the conclusion section. Please see in Line 389-391.
“Conclusion
In the mice model of colitis-induced liver injury, 2'-FL therapy increased GSH and T-AOC levels by 20.02% and 13.31%, respectively, and increased liver weight by 12.64% in comparison to DSS group. Furthermore, 2'-FL mediated in glucose and lipid related metabolism and alleviated liver oxygen stress through gut-liver-microbiotas axis. The above results provide a new perspective to understand the probiotic function of 2'-FL.”
We really appreciate your suggestions! Thank you so much for your help and considerations!
